# Role of Diatom Microstructure in Determining the Atterberg Limits of Fine-Grained Diatomaceous Soil

**Yiqing Xu [1,2], Xianwei Zhang [1,*], Gang Wang [1,2], Xinyu Liu [3] and Aiwu Yang [4]**

1   State Key Laboratory of Geomechanics and Geotechnical Engineering, Institute of Rock and Soil Mechanics, Chinese Academy of Sciences, Wuhan 430071, China
2   University of Chinese Academy of Sciences, Beijing 101408, China
3   School of Civil and Hydraulic Engineering, Huazhong University of Science and Technology, Wuhan 430074, China
4   School of Environmental Science and Engineering, Donghua University, Shanghai 200051, China
*   Correspondence: xwzhang@whrsm.ac.cn; Tel.: +86-138-8608-4677

**Abstract:** The presence of diatoms in diatomaceous soil gives it geotechnical properties that are unusual compared with common clays. The most typical physical property of diatomaceous soil is its abnormally high Atterberg limits compared to fine-grained soil without diatoms. For diatomaceous soil, the Atterberg limits are associated with many factors, such as diatom content, diatom crushing degree, etc. In the study reported here, it was ascertained that more diatoms lead to higher plastic and liquid limits. Once the diatoms are crushed, the plastic and liquid limits decrease. The pore fluid salt concentration barely influenced the Atterberg limits of diatomaceous soil. Additionally, the porous diatom microstructure and trimodal pore size distribution of diatomaceous soil were investigated via scanning electron microscopy, transmission electron microscopy, and mercury intrusion porosimetry. The underlying mechanism of abnormally high liquid and plastic limits of diatomaceous soil is revealed as the water stored in the special diatom microstructure. However, water in diatoms has no contribution to plasticity. Also discussed is the applicability of the current soil classification systems for diatomaceous soil. The findings of this study can help for a better understanding of Atterberg limits of diatomaceous soil and provide suggestions for the classification of diatomaceous soil.

**Keywords:** diatom; diatomaceous soil; Atterberg limits; microstructure; diatom crushing; soil classification

## 1. Introduction

Lacustrine and marine diatomaceous soil (DS) is found extensively in many countries, such as Mexico [1], Japan [2], Korea [3], Columbia [4], and China [5]. DS contains numerous fossilized diatoms with sizes in a range of several to dozens of micrometers [6] as a result of the accumulation of unicellular diatom cell walls (the diatom cell wall is referred to as frustules) [7]. A diatom frustule comprises mainly amorphous $SiO_2$ (Opal A) and some feldspar, quartz, pyrite, etc. [8–10]. The geotechnical characteristics of DS are significantly different from those of fine-grained soil without diatom; for example, (i) the porosity of natural DS can be as high as 74% [11], (ii) DS is characterized by a low specific gravity ($G_S$) of 2.26–2.38, which is lower than that of common clayey soil such as kaolin ($G_S$ = 2.775) and Singapore clay ($G_S$ = 2.770) [12], and (iii) natural DS has a high water content which can exceed 400% [13]. However, although DS has these relatively poor physical properties, it also has a high yield stress of ca. 540 kPa [14] and a high friction angle of 32–43° [15].

The Atterberg limits are among the most important physical properties of fine-grained soil and the plasticity index ($I_P$) is critical in soil classification. Many studies have investigated the plasticity properties of DS and provided a preliminary understanding of its Atterberg limits [6,12,16–18], showing that the Atterberg limits of DS are higher than the Atterberg limits of fine-grained soil without diatoms. For example, the liquid limit ($w_L$) and plastic limit ($w_P$) of artificial DS with a high content of pure diatoms can be as high

as 289.7% and 205.5%, respectively [19]; additionally, the $w_L$ value was found to be over 400% for natural DSs from Mexico and Columbia, and their $w_p$ value was ca. 80% [1,4]. Some studies showed that the Atterberg limits of DS are influenced by the mineralogy and fines fraction [12,20]. More diatoms may lead to a steady increase in Atterberg limits and the horizontal rightward proceeding across the A-Line in the plasticity chart [6]. Different diatom species also affect the plasticity properties of the DS; for example, DS contains disk-shaped diatoms that cause higher plastic and liquid limits than that contain tube-shaped diatoms [16].

Despite the current studies, more work is required to better understand the Atterberg limits of DS, i.e., (i) how the NaCl concentration of pore fluid and crushing degree of diatoms influence the Atterberg limits of DS (these two factors should be considered for DS because it conventionally forms in marine environments and contains either completely or partially crushed diatoms under the effect of geological burial), (ii) the microstructural difference between DS and common natural fine-grained deposits without diatom (also the association between the diatom microstructure and the Atterberg limits of DS and the corresponding underlying mechanism), and (iii) the applicability of the Unified Soil Classification System (USCS) for fine-grained DS, because this classification method involves the Atterberg limits and the particle size characteristics.

The study reported herein aimed to investigate the Atterberg limits of DS as affected by diatom content, diatom crushing, and pore fluid ionic concentration. For this purpose, the studied DS was obtained by mixing kaolin with pure diatoms of differing content and crushing degree, and NaCl solution replaced deionized water as the pore fluid for some of the diatomaceous samples. The plastic and liquid limits of this soil were measured by thread-rolling and fall-cone tests, the diatom and DS structures were studied microscopically via scanning electron microscopy (SEM) and transmission electron microscopy (TEM), and the pore characteristics of the DS were revealed via mercury intrusion porosimetry (MIP). Typical pores of the DS were analyzed quantitatively considering the diatom microstructural features. Next, the effects of the diatom content, diatom crushing degrees, and pore-fluid NaCl concentration on $w_L$, $w_p$, and $I_P$ were studied, thereby revealing the relationship between the microstructure and the plasticity properties of DS. Also discussed is the applicability of current classification methods (including USCS and pore fluid sensitivity classification methods) for fine-grained soil containing diatoms, based on the present investigations of Atterberg limits and particle size distribution (PSD). The present findings offer an improved understanding of the Atterberg limits of DS and insights into its classification.

## 2. Materials and Methods

### 2.1. Sample Preparation

The studied soil was a diatom–kaolin mixture (DKM) comprising Georgia kaolin (RP-2; Active Minerals International, Baltimore, MD, USA) and diatom from Changbai Prefecture, Jilin, China. Figure 1 gives the mineral and chemical compositions of kaolin and diatom as revealed by X-ray diffraction and X-ray energy-dispersive spectroscopy: diatom comprises quartz and opal with no clay minerals, which is consistent with the high contents of Si and O elements; kaolin has a dominant mineral content of kaolinite with traces of illite and quartz, in which the content of illite and quartz are ca. 1.8% and 1.2%, respectively. Table 1 summarizes the physical properties of the diatom and kaolin as determined by ASTM standards. Diatom has lower $G_S$ (2.25), significantly higher specific surface area (179.5 m$^2$/g), higher activity ($A_d$) (0.87), and cation exchange capacity (32.3 meq/100 g) compared with kaolin.

DKM samples with varying diatom proportions (0–100%) in terms of dry mass ratio were prepared to investigate the influence of diatom content on the consistency limits of fine-grained DS. To ensure that the water in diatom materials is completely removed before preparing DKM with a certain diatom content, the diatoms and kaolin materials were dried at 105 °C for 24 h. The DKM samples were prepared by thoroughly mixing

diatom and kaolin using mechanical stirring. For convenience, each prepared DKM sample was indexed as "diatom content D-kaolin content K", where D refers to the diatoms and K refers to the kaolin; for example, the DKM sample indexed as 40D-60K contained 40% diatoms and 60% kaolin.

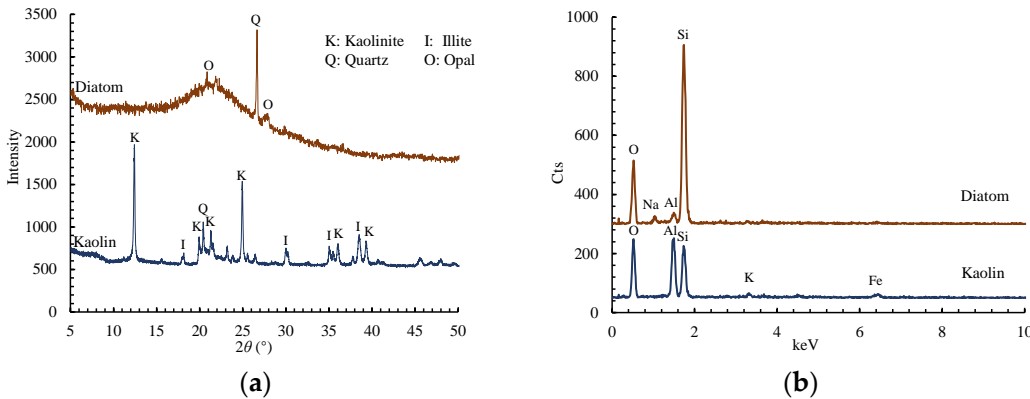

**Figure 1.** Mineral and chemical compositions of diatoms and kaolin as determined by (**a**) X-ray diffraction and (**b**) X-ray energy-dispersive spectroscopy.

**Table 1.** Physical properties of pure diatom and kaolin.

| Parameter | Value | |
|---|---|---|
| | **Diatom** | **Kaolin** |
| Specific gravity $G_S$ | 2.25 | 2.65 |
| Activity $A_d$ [1] | 0.87 | 0.26 |
| Average particle diameter $d_{50}$ (μm) | 6.80 | 0.59 |
| Specific surface area (m$^2$/g) [2] | 179.5 | 28.3 |
| Liquid limit $w_L$ (%) | 175 | 42 |
| Plastic limit $w_p$ (%) | 153 | 25 |
| Plastic index $I_p$ | 22 | 17 |
| USCS classification | MH | CL |
| pH | 8.11 | 7.26 |
| Cation exchange capacity (meq/100 g) | 32.3 | 9.8 |

[1] Activity [= ratio of plasticity index to a percentage by weight of clay-size particles]. [2] SSA was determined using the European methylene blue spot method [21].

### 2.2. Microstructure Observation

SEM and TEM observations were performed on the pure diatom and kaolin and their mixture to explore the microstructure characteristics of DS. The SEM observations involved a Quanta 250 (FEI, Eindhoven, The Netherlands) with an accelerating voltage of 20 kV. The samples were prepared via vacuum freeze-drying to avoid disturbing the microstructure during drying [22]. Samples were observed at the magnifications of 100×, 800×, 2000×, and 5000×, with particular attention paid to the interparticle linkages at the latter two magnifications. The TEM observations involved a field-emission TEM microscope (JEM 2100F; JEOL, Tokyo, Japan) operated at 200 kV. The samples were prepared by dropping a suspension of diatoms and kaolin onto a carbon-coated copper grid; the suspension was obtained by dispersing dry diatom and kaolin particles in deionized water.

Additionally, MIP tests were performed to reveal the pore size distribution of the DKM. DKM slurry was pre-consolidated at 60 kPa, vacuum freeze-dried, and cut into 5 mm × 10 mm × 5 mm pieces to prepare a sample for an MIP test. The MIP tests involved a Quantachrome Pore Master 33, which could detect pores with a diameter in the range of 0.006–300 μm, and they were based on the equation proposed by Washburn [23]:

$$P = -\frac{4\sigma_{Hg}\cos\theta_{Hg}}{d} \tag{1}$$

which describes the correlation between the applied pressure $P$ and the equivalent pore diameter $d$. Here, $\theta_{Hg}$ is the mercury–soil contact angle, and $\sigma_{Hg}$ is the mercury–soil surface tension; the values used in this study were 141.3° and 0.48 N/m, respectively. Using the pore diameter $d_i$ calculated from this equation, the cumulative pore volume (*CV*) and the log-incremental pore volume (*IV*) are calculated as:

$$CV = \sum_i \Delta V_i \tag{2}$$

$$IV = \frac{-\Delta V_i}{\Delta \log d_i} \tag{3}$$

The MIP results are given as cumulative and log-incremental pore volume distribution curves, i.e., *CV*–*d* and *IV*–*d* curves.

### 2.3. Details of Tests for Atterberg Limits

In this study, the $w_p$ of the soil was determined using thread-rolling tests following ASTM D4318-17 [24], while the $w_L$ was determined by fall-cone tests, which give less variable experimental results compared to the standard "Casagrande cup" method [25–27]. Note that a thread-rolling test was inappropriate for sample 100D-0K because of the non-plastic nature of pure diatoms (Figure 2); therefore, the method proposed by Vardanega et al. [28] was used to determine $w_p$ for pure diatoms from the correlation between $I_P$ and the flow index of the fall-cone test. Additionally, the PSD curves of the prepared DKM samples were obtained to determine their clay and silt fractions; for this, the hydrometer method [29] was used because the DKM samples contained no coarse particles (>0.075 mm).

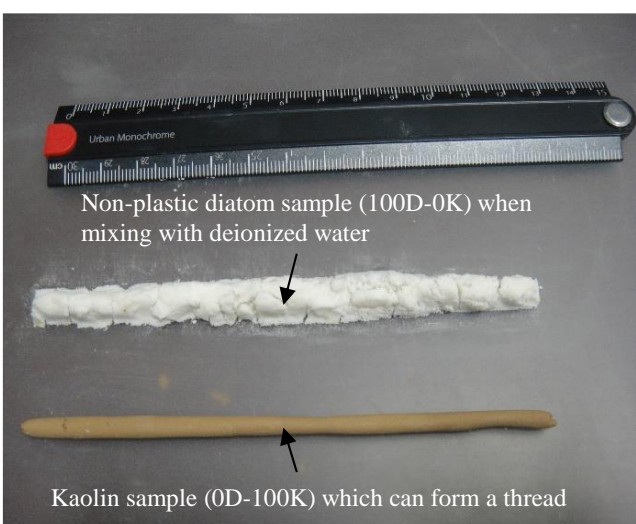

**Figure 2.** The successful case of kaolin and failed attempt of diatoms to form a thread.

To study the effect of the pore fluid ionic concentration on the Atterberg limits of diatom-rich soil, NaCl solutions with a concentration of 0.01 M, 0.1 M, 1 M, and 2 M were prepared. These NaCl solutions are used to replace deionized water as the pore fluid of DKM. Additionally, kerosene was used as a pore fluid to determine the liquid limit $w_{L,ker}$. The Atterberg limits of these DKMs with their original pore fluids (deionized water) replaced by NaCl solutions and kerosene were measured through thread-rolling and fall-cone tests.

As well as the NaCl concentration of pore fluid, the crushing of diatoms was also considered because the diatoms in natural DS are usually crushed to various degrees. To prepare a DKM sample with crushed diatoms (hereinafter referred to as crushed DKM, e.g., crushed 80D-20K), the ball milling method was performed to obtain crushed diatoms: dry raw diatoms were sent to a planetary ball mill (FRITSCH pulverisette; Shanghai, China) and were mechanically stirred in an agate vessel (500 mL) filled with hundreds of corundum

grinding balls (8 mm in diameter) and operated at a rotational speed of 100 rpm; diatoms with various crushing degrees were obtained by changing the milling duration (1 h, 2 h, and 4 h), with 4 h expected to be sufficient to crush diatoms completely. The crushed DKM sample was then prepared by mixing the crushed diatoms as in the case of uncrushed DKM (see Section 2.1). The $w_L$ and $w_p$ values of the crushed DKM samples were also measured using thread-rolling and fall-cone tests and the complete crushed diatoms (with milling duration of 4 h) were observed using SEM to examine their altered microstructure due to diatom crushing.

## 3. Results

### 3.1. Microstructure of DKM

#### 3.1.1. Overview of DKM Microstructure

Figure 3 shows the DKM microstructure as revealed by SEM images. Kaolin aggregates are observed in the form of layered kaolinite platelets and the kaolinite platelets are associated with each other via face-to-face, edge-to-face, and edge-to-edge interparticle contacts (Figure 3a). The diatoms are mainly intact disk-shaped ones (*Stephanodiscus* and *Cyclotella*) and their fragments (Figure 3b): the diameters of intact *Cyclotella* and *Stephanodiscus* diatoms are ca. 70 μm and ca. 30 μm, respectively, whereas the size of a smallest diatom fragment is equal to that of a kaolin platelet; also presented in this figure are some tube-shaped diatoms (*Aulacoseira*, with a diameter of ca. 10 μm). Figure 3d–f shows the typical microstructural features of DKM samples. The kaolinite platelets and diatoms gathered together to form the diatom–kaolin aggregate, which has massive pores within it. According to Tanaka and Locat [2], the pore types of DKM include inter-aggregate, intra-aggregate, skeletal, and intra-skeletal: the former two are due to kaolin aggregates, whereas the latter two are due to diatom frustules. For kaolin, the inter-aggregate and intra-aggregate pores of various sizes are identified in Figure 3a. For the diatom, the skeletal pores with a diameter of 0.1–1 μm and an intra-skeletal pore can be seen in the magnified image of a single disk-shaped diatom (Figure 3c). For DKM, all types of pore are found in diatom–kaolin aggregates (Figure 3f); some skeletal pores therein are surrounded or even filled by kaolinite clay particles, which provide a connection between the intact diatoms and diatom fragments. Figure 3g–i shows SEM images of a chain-like skeleton structure of crushed diatoms, which is formed by the 15–25 μm long fragments of diatom frustules. Figure 3g,h shows that a few tiny frustules with diameters of 12–25 μm remain intact. Additionally, the intact diatoms have dominant intra-skeletal and skeletal pores, but these types of pores mostly disappear in the crushed diatoms because the skeletal structure of the diatom has been damaged, leaving the inter-aggregate and intra-aggregate pores to dominate.

TEM images of the kaolin and diatoms are shown in Figure 4. The kaolinite platelets with a size of several hundred nanometers are stacked mainly via face-to-face contacts (Figure 4a,b). Also visible in these images is the tube-like halloysite formed by rolled hydrate kaolinite flakes; the halloysite occasionally appeared along with the kaolinite platelets and interspersed among them. Figure 4c shows a two-dimensional image of the internal structure of a 3.5 μm diameter diatom particle; inter-connective pores are uniformly and hexagonally arrayed on the profile surface of the frustule and the TEM image with magnification of 80,000× (Figure 4d) shows that the diameter of the pores observed in the hexagonal arrangement from Figure 4c is ca. 200 nm.

#### 3.1.2. Pore Characteristics of DKM

The pore size distribution curve of pure kaolin is characterized by a single prominent peak at diameter of 0.2–0.6 μm (Figure 5a), which represents the inter-aggregate pores according to Figure 3. The domain of intra-aggregate pores was not found in the kaolin sample because this type of pore is not abundant, resulting from the dominant face-to-face contact between kaolinites (Figures 3 and 4). For diatom, the dominant domain corresponding to diameters of 0.06–0.2 μm and the subdomain at diameter of 10 μm

represent the skeletal and intra-skeletal pores, respectively. Skeletal pores are small in volume but numerous, which is opposite to the case of intra-skeletal pores. The pore size distribution curve of diatoms in the range of $d = 0.2–0.6\ \mu m$ is more like a plateau than a peak. This plateau, as well as the second-highest peak ($d = 0.6–2\ \mu m$), correspond to the pores in the open fabric of the diatom skeletons (Figure 3b). The pore size distribution of DKM is trimodal and located between those of kaolin and diatoms. For DKM, the domain for $d = 0.2–2\ \mu m$ corresponds to the intra-aggregate pores, inter-aggregate pores, and the pores in the diatom skeletons. The skeletal and intra-skeletal pores in the DKM are reflected by the peaks at diameters of $0.06–0.2\ \mu m$ and $10\ \mu m$, respectively, which is consistent with the case for the diatoms. With increasing diatom content, the pore size distribution curve of the DKM approaches that of the diatoms in shape.

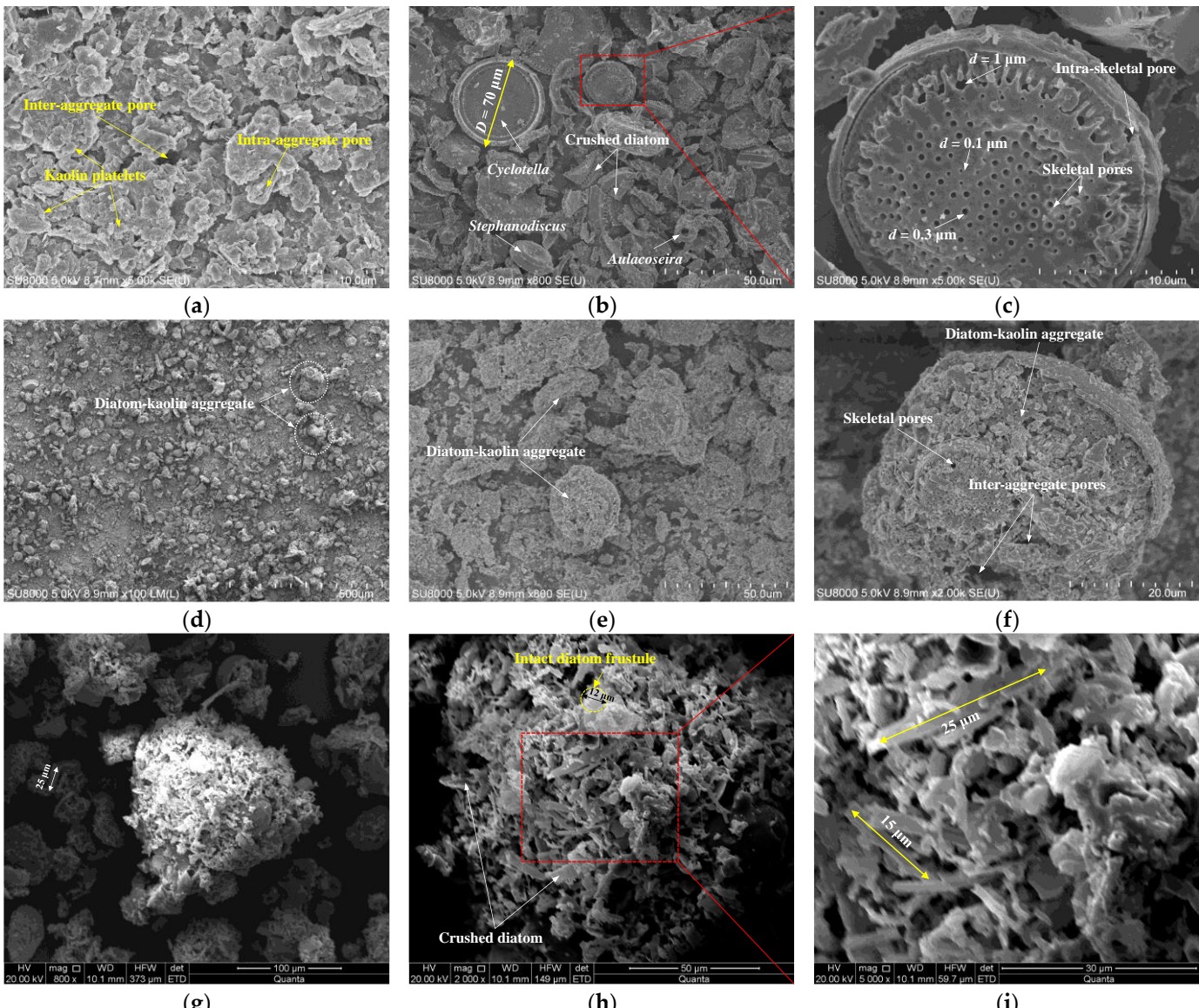

**Figure 3.** Scanning electron microscopy images of (**a**) kaolin (0D-100K) (**b**,**c**) diatoms (100D-0K), (**d**–**f**) a DKM (60D-40K), and (**g**–**i**) crushed diatoms.

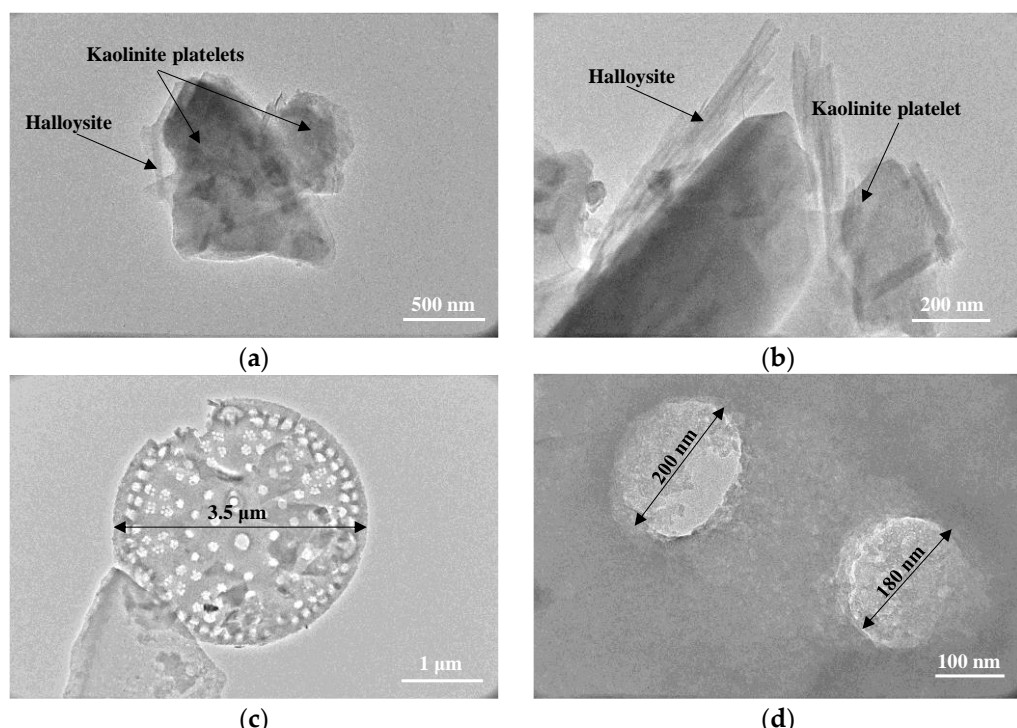

**Figure 4.** Transmission electron microscopy images of (**a**) kaolin (15,000× magnification); (**b**) kaolin (40,000×); (**c**) diatoms (8000×); and (**d**) diatoms (80,000×).

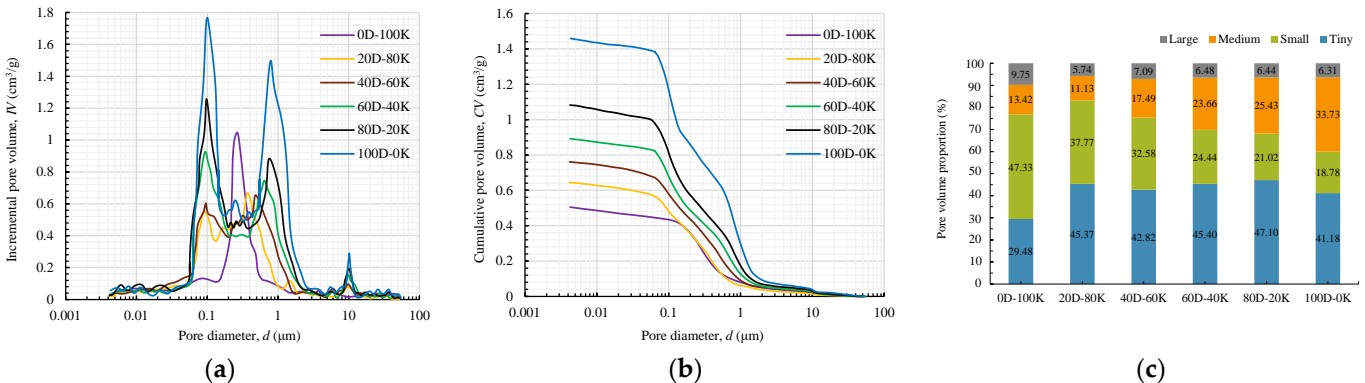

**Figure 5.** Results of mercury intrusion porosimetry tests on DKM samples: (**a**) *IV*–*d* curves; (**b**) *CV*–*d* curves; (**c**) volume proportion of pores with different diameters.

Figure 5b shows the cumulative pore distribution of DKM, which is intermediate to that of diatom and kaolin. The *CV*–*d* curve of kaolin falls rapidly as *d* increases from 0.2 μm to 0.6 μm, while that of diatom falls in a stepwise manner as *d* increases from 0.06 μm to 2 μm; the *CV*–*d* curves of both kaolin and diatom for other pore ranges are approximately unchanged. As the diatom proportion increases, the pore volume considerably increases, and the pattern of the *CV*–*d* curve progressively changes from the kaolin to the diatom.

Based on the *IV*–*d* curves, the DKM pores are classified as tiny (*d* < 0.2 μm), small (0.2 < *d* < 0.6 μm), medium (0.6 < *d* < 2 μm), and large (*d* > 2 μm) following the study of Zhang et al. [30]. The diatoms have more medium and tiny pores, but far fewer small pores compared to the kaolin (Figure 5c). When increasing the diatom content in DKM, the number of medium pores and small pores increases, which corresponds to the rightward shift of the second dominant peak (Figure 5a). By contrast, the proportion of tiny pores is scarcely affected by the diatom content and is higher than that of the kaolin. From a micro-level perspective (Figure 3a,f), the difference between the pore volume distribution curves of kaolin and DKMs in Figure 5c is due mainly to the skeletal and intra-skeletal

pores from the diatom frustules and the varying sizes and shapes of inter-aggregate and intra-aggregate pores in diatom–kaolin aggregates.

### 3.2. Atterberg Limits of DKM

#### 3.2.1. Effect of Diatom Content

The Atterberg limits of DKM samples with varying diatom contents are presented in Figure 6, along with the values of $w_L$, $w_p$, and $I_P$ for different DKMs from previous studies [6,12,31–34]. The more diatoms in the DKM, the higher $w_L$ and $w_p$ are. The Atterberg limits of DKM with a diatom content of 0–60% are lower than DKM with a diatom content of 80–100%; for example, sample 60D-40K has $w_L = 70\%$ and $w_p = 46\%$, while sample 80D-20K has the higher values of $w_L = 128\%$ and $w_p = 98\%$. However, although $w_L$ and $w_p$ of the DKM change significantly with diatom content, $I_P$ remains relatively stable because the increments in $w_L$ and $w_p$ are approximately the same. Notably, $w_L$ and $w_p$ increase remarkably when the diatom content increases from 60% to 80%. Some previous studies [12,35] suggested that the correlation between increasing $w_L$ and $w_p$ and diatom content is because $w_L$ and $w_p$ are affected by the unique microstructure of diatom frustules.

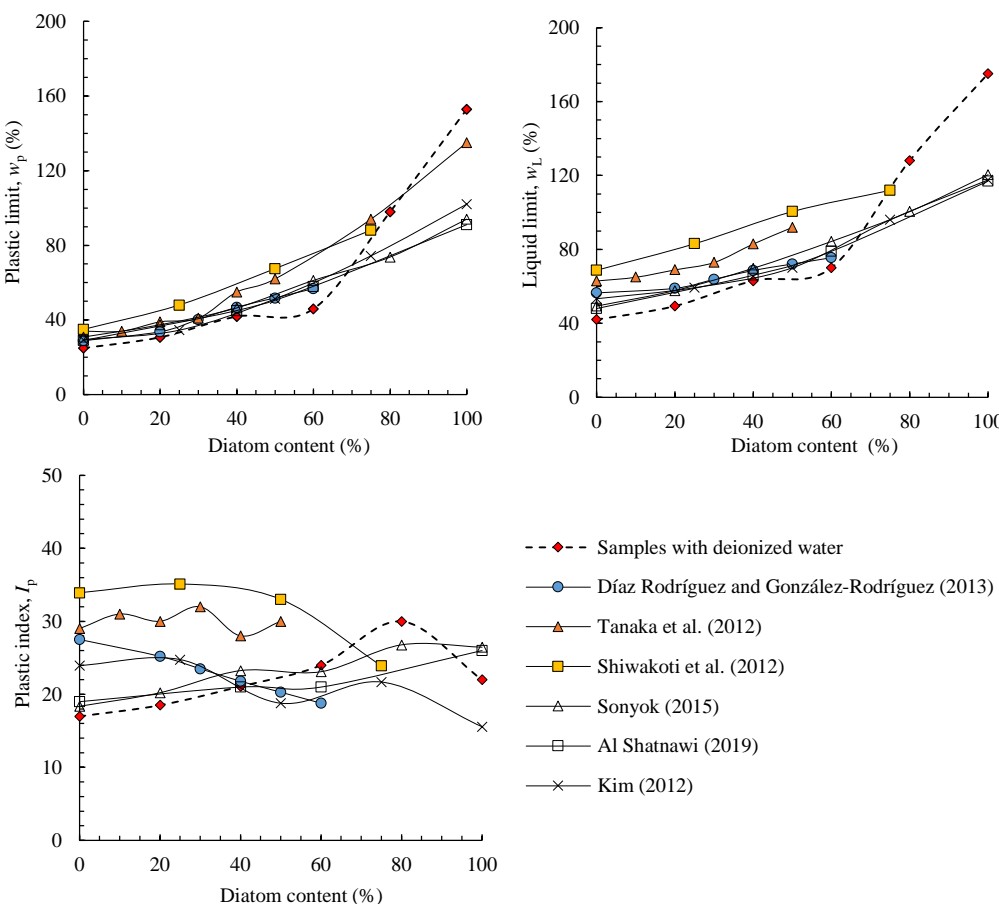

**Figure 6.** Values of $w_p$, $w_L$, and $I_P$ for DKM samples versus diatom content. Data sources: DKMs [6,12,31–34].

The relationships between $I_P$ and $w_L$ for various DKMs are shown in Figure 7 as a Casagrande plasticity chart, along with data points for natural diatomaceous sediments from China [5], Korea [3], Japan [2], and Columbia [4]. The data points in Figure 7 for DKM samples move rightward with increasing diatom content, indicating that the plasticity of a DKM is hardly affected by its diatom content. Additionally, all of the DKM data points fall below the A-line, whereas the data of natural DSs are above the A-line and approach the U-line, as shown by the colored areas in Figure 7. The higher positions of the data

points for natural DS compared with those for artificial DKM samples may be due to the existence of high plasticity mineral montmorillonite [1,35]. Figure 8 shows the relationships between $I_P$ and clay content for DKMs and natural DSs, where the slope of the curve represents soil activity $A_d$. The $A_d$ value increases with the clay fraction and the DKM data points change from the line of $A_d = 0.5$ to $A_d = 1$, which is consistent with the findings of Shiwakoti et al. [12].

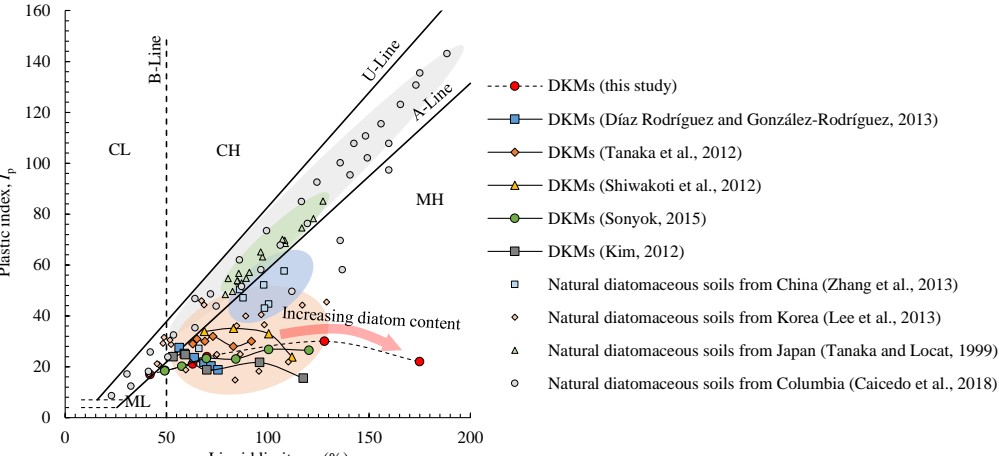

**Figure 7.** Casagrande plasticity chart for diatomaceous soil (DS). Data sources: DKMs [6,12,31,32,34], Natural diatomaceous soils [2–5].

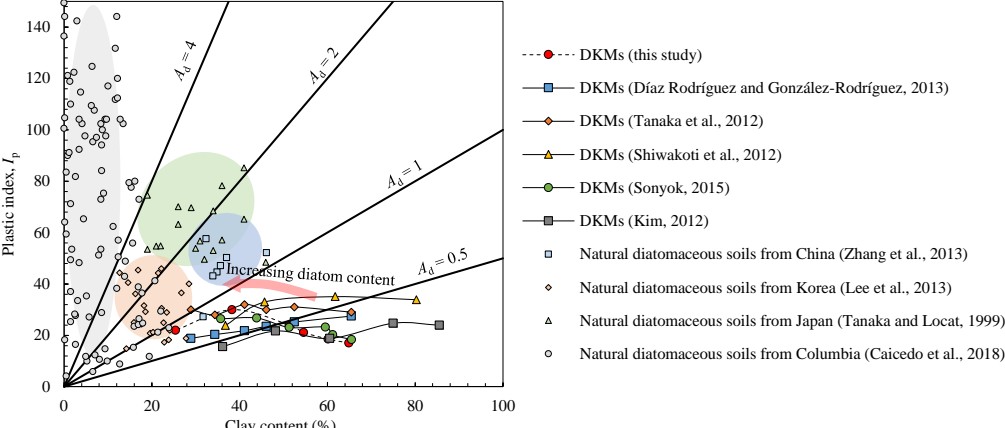

**Figure 8.** Relationship between $I_P$ and clay fraction for different DKM samples. Data sources: DKMs [6,12,31,32,34], Natural diatomaceous soils [2–5].

### 3.2.2. Effect of Diatom Crushing

Figure 9 shows how diatom crushing affects the Atterberg limits of the DKM samples. The crushed DKM samples have lower $w_L$ and $w_p$ values than uncrushed ones. In particular, this difference in $w_L$ and $w_p$ between crushed and uncrushed DKMs is more remarkable for samples 80D-20K and 100D-0K. The $w_L$ values of crushed 80D-20K (58.4%) and 100D-0K (62.4%) are much lower than those of uncrushed 80D-20K (128%) and 100D-0K (175%). This pattern of $w_L$ is similar to $w_p$. The results mean that the high $w_L$ and $w_p$ values for samples 80D-20K and 100D-0K are due mainly to the intact microstructure of the diatoms. As shown in Figure 3h,i, crushed diatom fragments form a chain-like structure that cannot store water, because the water in the crushed diatom can only be stored in the skeletal pores of the fragments and not in the intra-skeletal pores.

Naturally, milling for longer leads to further fragmentation of diatom frustules and lower $w_L$ and $w_p$ (Figure 9) because crushing intact diatoms causes collapsed intra-skeletal pores and reduced water retention capacity. Nevertheless, the $I_P$ values for the DKM samples were scarcely affected by diatom crushing, meaning that the water stored in the intra-skeletal pores of intact diatom frustules contributes little to the plasticity of the DS.

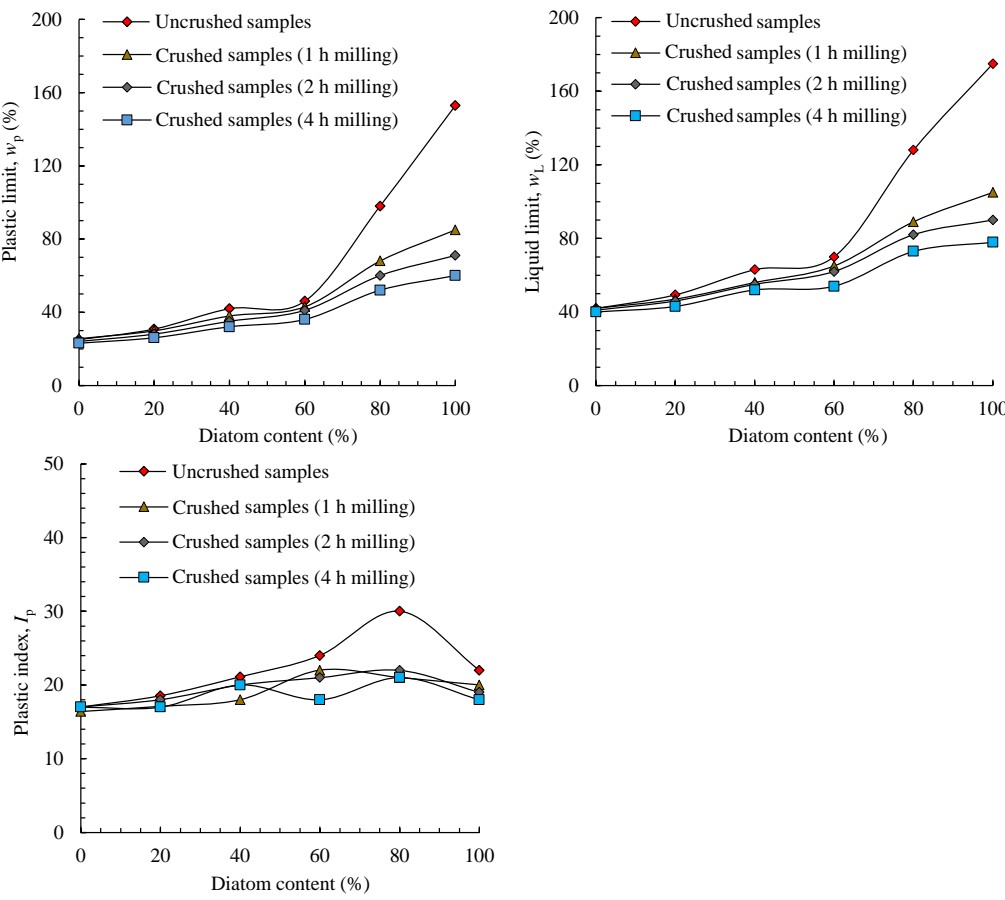

**Figure 9.** Values of $w_p$, $w_L$, and $I_P$ for DKM samples with different diatom contents.

### 3.2.3. Effect of Pore Fluid Ionic Concentration

Figure 10 shows the test results for the Atterberg limits of the DKM samples with different values of NaCl concentration. For the DKM samples, $w_L$, $w_p$, and $I_P$ scarcely change with increasing NaCl concentration; however, for pure kaolin, $w_L$, and $w_p$ increase initially with increasing NaCl concentration and then decrease when the latter reaches 0.1 mol/L, while $I_P$ tends to decrease with increasing NaCl concentration. Similar conclusions for kaolin were reached by Palomino and Santamarina [36], who reported that NaCl concentration significantly affects the interparticle association of the kaolinite fabric, thereby affecting the Atterberg limits. For kaolin with a high ionic concentration (>0.1 mol/L), face-to-face contacts dominate and the double layer is thin, leading to low $w_L$ and $w_p$; this effect of NaCl concentration on the kaolinite fabric weakens as more diatoms are added to the kaolin [36]. Although the Atterberg limits of DKM are insensitive to the ionic concentration, they increase apparently with increasing diatom content; this indicates that diatom content has far more influence on the Atterberg limits than does ionic concentration, which is consistent with the findings of Palomino et al. [37].

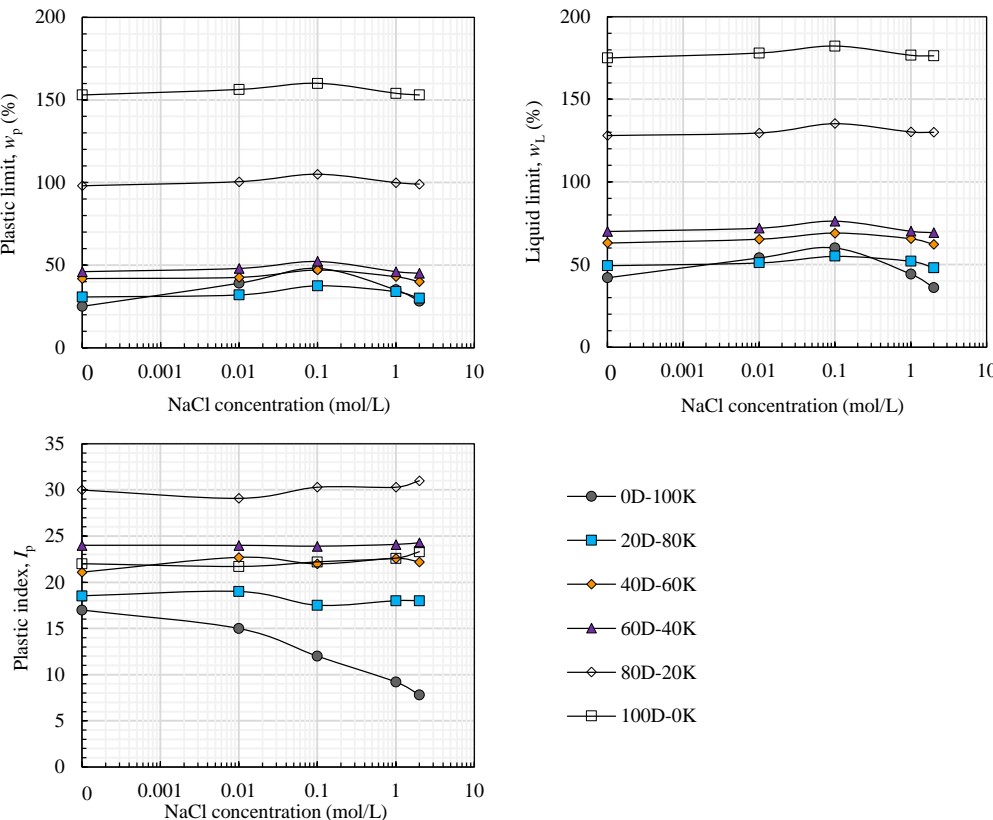

**Figure 10.** Values of $w_p$, $w_L$, and $I_P$ for DKM samples with different NaCl concentrations.

## 4. Discussion

### 4.1. Influence of Diatoms on Atterberg Limits from a Microstructural Perspective

The diatom and kaolin structures are shown schematically in Figure 11. A disk-shaped diatom is a spheroid cell with two symmetrical porous shells connected by a girdle band. The shells have a regular and centrosymmetric arrangement of skeletal pores on their surfaces. Inside the spheroid cell is a large inner chamber that stores the biological protoplasm of the diatom; this chamber becomes a typical intra-skeletal pore in a frustule after the diatom dies. Furthermore, Losic et al. [38] showed that the disk-shaped diatom frustule comprises three membranes at the nanoscale—namely the areola, cribium, and cribellum—containing large, small, and tiny nanoholes, respectively. Figure 11b shows the distribution of nanoholes in cribellum and cribium corresponding to a large nanohole in the areola membrane. Notably, the pores in the cribellum were observed in the TEM images (Figure 4c,d) from the vertical direction of a frustule, as shown by the arrow in Figure 11b. As described above, the microstructure of the diatom frustule is porous and water is stored in the skeletal and intra-skeletal pores. These pores are connected through the nanoholes inside the frustule so that the $w_L$ and $w_p$ values and water content of DS are measured high. Figure 11c shows the crystal structure of kaolinite with two tetrahedral and octahedral sheets. The structure of kaolin clays is a layered stacking with limited space inside, and the water in kaolin is adsorbed in the surface of kaolinite platelets in the form of a double layer, which is different from the case of the diatom. However, although the higher consistency limits of diatomaceous samples compared to kaolin caused by the water inside the diatom frustules were measured, the kaolin and diatomaceous samples have similar $I_P$ values. This is because the water inside the diatoms contributes little to the plasticity of the soil. Furthermore, it is observed that the diatoms with $I_P = 22$ are non-plastic, whereas the kaolin with $I_P = 17$ is plastic; therefore, the applicability of $I_P$ as a parameter for soil plasticity evaluation and soil classification for DS should be assessed carefully, especially when the DS contains 20% or more diatom microfossils (by weight).

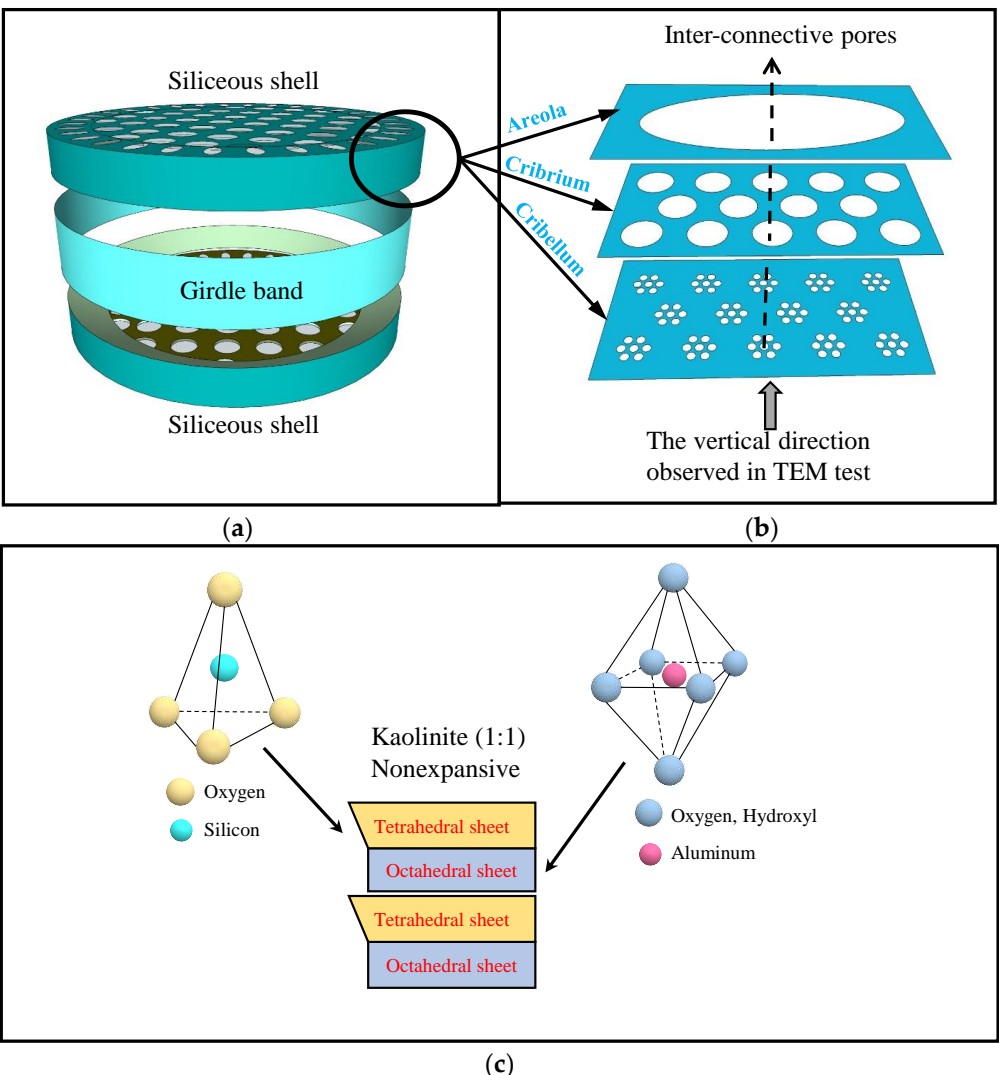

**Figure 11.** Schematics of (**a**) disk-shaped diatom frustule, (**b**) profile structure of frustule, and (**c**) crystal structure of kaolinite.

Apart from the diatom content, the Atterberg limits of DS are also related to diatom crushing. The crushed DKM samples have lower $w_L$ and $w_P$ values than uncrushed ones. This is because the hollow intra-skeletal and skeletal pores collapsed, thereby reducing water retention capacity, and the higher crushing degree of diatoms leads to the lower Atterberg limits. Moreover, the presence of diatoms affects the response of the Atterberg limits of the soil to the NaCl concentration. The $w_L$, $w_P$, and $I_P$ values of the DS hardly changed upon changing the NaCl concentration, whereas those of the kaolin did (Figure 10). For pure kaolin, different values of NaCl concentration lead to various particle associations among kaolinites and further changes in the forces among clay particles [36]. For pure diatoms, the double layer is thin because of the low surface charge density of a frustule and does not change with NaCl concentration [20,39]. The water that significantly affects the consistency limits of the diatoms is stored mainly in the intra-skeletal pores instead of in the double layer. Interestingly, the effects of NaCl concentration on the kaolin sample almost disappear with the addition of diatoms (Figure 10), which may be because of the unique interactions between kaolin and diatom particles of the diatom–kaolin aggregates.

### 4.2. Classification of Fine-Grained Soil Containing Diatoms

Before classifying the DKM samples, their particle size compositions are obtained from the PSD curves (Figure 12). The diatoms comprise mainly silt, while the kaolin contains more clay than silt. The PSD curves of the DKM samples move downward with increasing diatom content. Higher diatom content in the DKM corresponds to more silt and less clay (Figure 13), and DKM with a diatom content of more than 60% has a dominant composition of silt and is known as silt-dominated soil. Figure 14 shows that the $d_{50}$ value of the DKM increases with increasing diatom content: $d_{50}$ increases by 131% when the diatom proportion changes from 60% to 80%. Similar phenomena have been observed in previous studies [6,32], except for that by Wiemer and Kopf [19], who used clayey silt as a substitute for kaolin to prepare the studied DS.

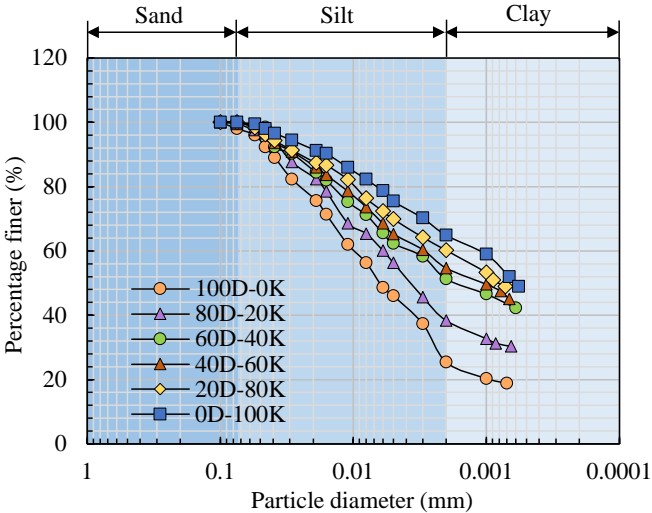

**Figure 12.** Particle size distribution of DS with different diatom content.

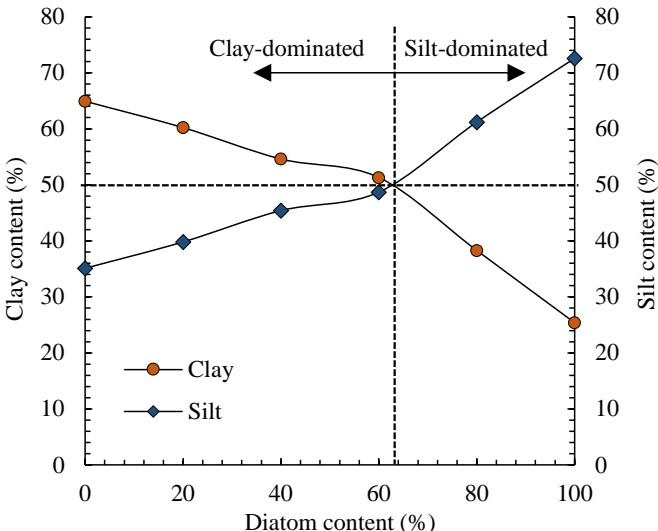

**Figure 13.** Variations of silt and clay particle contents of DKM samples with diatom content.

Following the USCS (Figure 7), samples 0D-100K, 20D-80K, and the others were classified as low-plasticity clay (CL), low-plasticity silt (ML), and high-plasticity silt (MH), respectively. According to the Casagrande plasticity chart, the rightward shift of the DKM data points with increasing diatom content indicates the change of soil behavior from clayey to silty, and the DKM samples with $w_{L} > 50\%$ are of high plasticity. Similar tendencies for types of DKM in other studies can be seen in Figure 7. Considering the classification results, the transition of dominant particle size from clay to silt with increasing diatom content for

the DKM samples is confirmed by the PSD curves in Figures 12 and 13. However, the results for high-plasticity soil are inappropriate according to the investigation of the microstructure and consistency limits of the DKM samples. For example, sample 100D-0K is non-plastic and should not be classified as soil with high plasticity (Figure 2). Additionally, sample 20D-80K contains dominant clay-sized particles but is misclassified as silt by the USCS. Therefore, the USCS may well result in improper classification of DS because of its unique consistency limits due to the diatom microstructure.

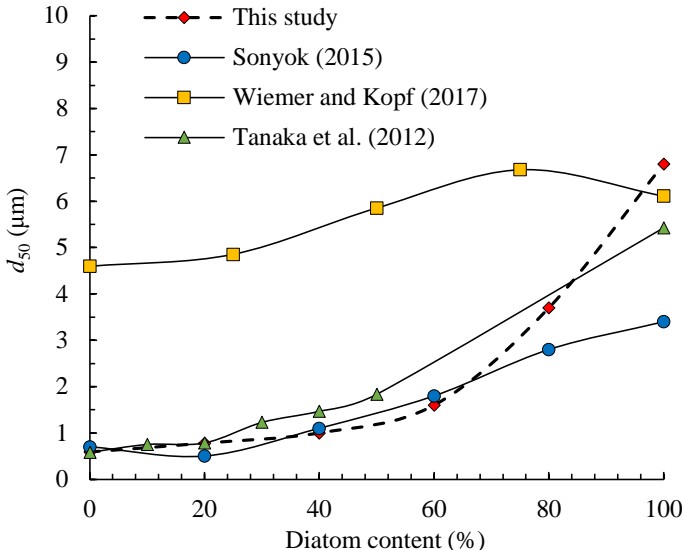

**Figure 14.** Relationship between average particle diameter $d_{50}$ and diatom content. Data sources: DKMs [6,19,32].

Jang and Santamarina [27] proposed a classification system for fine-grained soil based on pore fluid sensitivity. This classification focuses on the different types of interparticle interaction (i.e., van der Waals and double-layer effects) in fine-grained soil. It uses electrical sensitivity ($S_E$) to measure the permittivity and conductivity of the soil. $S_E$ is calculated using the fall-cone liquid limits, which are measured with different pore fluids of deionized water, 2-M NaCl solution, and kerosene (i.e., $w_{L,DW}$, $w_{L,brine}$, and $w_{L,ker}$, respectively). Figure 15 shows the classification results according to this system for different types of DKM and the DS studied by Al Shatnawi and Bandini [7], along with those for silica silt, kaolin, and other minerals (i.e., mica, bentonite, and illite). With increasing diatom proportion, the permittivity decreases, and the conductivity increases slightly, which results in decreasing $S_E$ (Figure 15a). As shown in Figure 15b, a DKM with a diatom content of less than 40% is classified as "intermediate plasticity, intermediate $S_E$", while one with a diatom content of more than 80% is classified as "high plasticity, low $S_E$". Sample 60D-40K is classified as "high plasticity, intermediate $S_E$". However, the pore fluid sensitivity classification recognizes pure diatoms as high plasticity soil, although this sample is confirmed to be non-plastic (Figure 2).

Shiwakoti et al. [12] showed that the silt-sized diatom particles in soil are, in effect, like clay particles, leading to a high value of $A_d$. This is also the case in the present study, and Figure 8 shows that adding diatoms to kaolin results in a higher value of $A_d$. However, unlike clay particles, diatoms are non-plastic because their frustules are connected through a chain-like skeleton structure (Figure 3i). In this case, the consistency limits of DS are essentially different from those of fine-grained soil without diatoms. Consequently, further investigations are necessary to propose a more reasonable classification system for DS, considering the unique microstructure of diatom frustules.

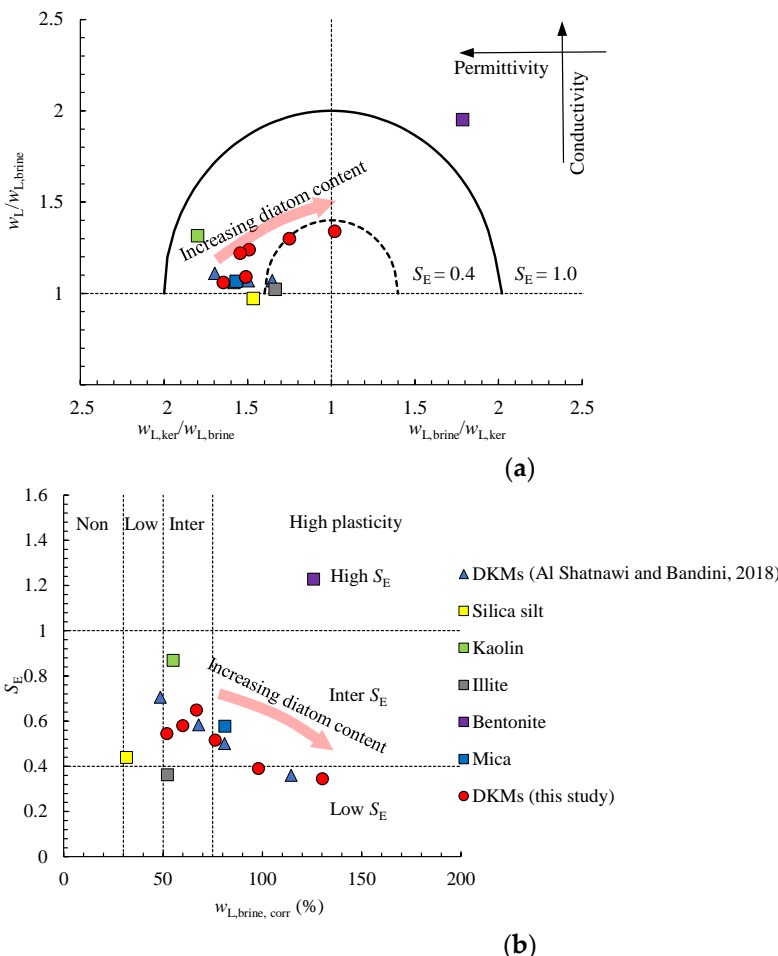

**Figure 15.** Classification of DS according to pore fluid sensitivity: (**a**) permittivity and conductivity of the samples; (**b**) electrical sensitivity of the samples. Data sources: DKMs [7].

## 5. Summary and Conclusions

This study investigated how diatom content, diatom crushing degree, and pore fluid ionic concentration influence the consistency limits of DS, along with the underlying microscopic mechanism. The main conclusions are as follows:

Diatom frustules have a porous microstructure with massive skeletal and intra-skeletal pores, which provide space to store water. The crushed diatom fragments form chain-like connective skeletons and have no intra-skeletal pores. The microstructure of a DKM is characterized by diatom–kaolin aggregates, with the pores of the diatom frustules surrounded or filled by clay particles.

The pore size distribution curve of kaolin changes from unimodal to trimodal with increasing diatom content because of the introduction of skeletal and intra-skeletal pores. With increasing diatom content in a DKM, the volume proportion of tiny and medium pores increases, and that of small pores decreases, which is closer to the pore characteristics of diatoms than those of kaolin.

A DKM with a diatom content of over 60% is silt-dominated; otherwise, it is clay-dominated. With an increasing diatom proportion, the clay fraction of a DKM decreases approximately linearly, but its activity increases with the clay fraction. The diatom frustules are active silt-sized particles, in effect, like clay particles in the soil.

The $w_L$ and $w_P$ values increase significantly with increasing diatom content, but $I_P$ remains almost constant. The extremely high $w_L$ and $w_P$ values of DS are associated with the intact diatom frustules: the $w_L$ and $w_P$ values of a crushed DKM are lower than those of an uncrushed one because the collapsed intra-skeletal pores in the former can no longer

retain water; however, the water stored in the intra-skeletal pores of diatom frustules contributes little to the plasticity of the soil.

The Atterberg limits of kaolin are affected by NaCl concentration, whereas those of pure diatoms are almost unaffected. Similar to the consistency limits of pure diatoms, the $w_L$, $w_p$, and $I_P$ values for a DKM with even a small proportion of diatoms (20%) are insensitive to NaCl concentration.

The plasticity and pore fluid sensitivity charts fail to classify DS accurately, especially when the diatom content is high. Further investigations are needed to establish a more reasonable classification system for DS, considering its unique microstructure and diatom content.

This study considered the effect of geological burial in the marine environment during the forming process of DS by investigating the influences of diatom content, diatom crushing degree, and pore fluid ionic concentration on Atterberg limits. Additionally, the microstructure and pore characteristics of DS were examined via microscopic tests. Through the test results of Atterberg limits and microstructure of DS, this study revealed the micro-level mechanism of the high Atterberg limits of DS. In this study, only disk-shaped diatoms are considered, which means the results of this work may not apply to DS dominating other diatom species. More diatom species and other factors, such as complex mineral components of natural DS, should be considered in further research.

**Author Contributions:** Conceptualization, X.Z. and X.L.; Methodology, Y.X., X.L. and A.Y.; Validation, Y.X.; Formal analysis, Y.X., G.W. and X.L.; Writing—original draft, Y.X. and X.L.; Writing—review & editing, X.Z. and G.W.; Supervision, X.Z.; Project administration, X.Z.; Funding acquisition, X.Z. All authors have read and agreed to the published version of the manuscript.

**Funding:** This work was supported by the National Natural Science Foundation of China (Nos. 41972285, 42177148, 41972293), the Youth Innovation Promotion Association CAS (No. 2018363), and Science Fund for Distinguished Young Scholars of Hubei Province (Grant No. 2020CFA103). The APC was funded by the National Natural Science Foundation of China (No. 41972285).

**Data Availability Statement:** All data that supports the findings of this study are available from the corresponding author upon reasonable request.

**Acknowledgments:** The author would like to thank Fei Wang of China Railway Design Corporation (Tianjin, China) for his invaluable help in methodology. The financial support of the National Natural Science Foundation of China (Nos. 41972285, 42177148, 41972293), the Youth Innovation Promotion Association CAS (No. 2018363), and Science Fund for Distinguished Young Scholars of Hubei Province (Grant No. 2020CFA103) are gratefully thanked.

**Conflicts of Interest:** The authors declare no conflict of interest.

## Nomenclature

| | |
|---|---|
| $G_S$ | Specific gravity |
| DS | Diatomaceous soil |
| $I_P$ | Plasticity index |
| $d_{50}$ | Average particle diameter (μm) |
| $w_L$ | Liquid limit (%) |
| $w_p$ | Plastic limit (%) |
| USCS | Unified Soil Classification System |
| SEM | Scanning electron microscopy |
| TEM | Transmission electron microscopy |
| MIP | Mercury intrusion porosimetry |
| DKM | Diatom-kaolin mixture |
| $A_d$ | Activity |
| D | Diatom |
| K | Kaolin |
| $P$ | Applied pressure (Pa) |
| $d$ | Equivalent pore diameter (m) |

| | |
|---|---|
| $\theta_{Hg}$ | Mercury–soil contact angle (°) |
| $\sigma_{Hg}$ | Mercury–soil surface tension (N/m) |
| *CV* | Cumulative pore volume |
| *IV* | Log-incremental pore volume |
| PSD | Particle size distribution |
| CL | Low-plasticity clay |
| ML | Low-plasticity silt |
| MH | High-plasticity silt |
| $w_{L,ker}$ | Liquid limit using kerosene (%) |
| $w_{L,brine}$ | Liquid limit using NaCl concentration (%) |
| $S_E$ | Electrical sensitivity |

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
