# Peer review of "Role of Diatom Microstructure in Determining the Atterberg Limits of Fine-Grained Diatomaceous Soil"

_applsci, doi:10.3390/app13042287_

Round 1

Reviewer 1 Report

 Role of diatom microstructure in determining the Atterberg limits of fine-grained diatomaceous soil

The manuscript focuses on studying the changes of some geotechnical properties in diatomaceous soils, according to diatom content, diatom crushing, and pore-fluid ionic concentration.

The work is very interesting and very well structured. Presents new data and appropriately reviews and references existing literature on the topic. The tables and figures are also adequate and understandable. In turn, the methodology is described correctly, and it is worth mentioning that the analytical work is scrupulous and was carried out considering various ASTM standards. The reading is fluent.

That is why the acceptance of the manuscript is recommended, and only minor changes are suggested, as detailed below:

1. XRD data of the kaolinite sample (figure 1a) should be reported in semi-quantitative form. Check the presence of illite in the sample because the 001 peak of this phase is not observed in the diffractogram of said figure.

2. It is important to specify what type of opal (Opal A) occurs in diatomites (for example: Ghisoli, C., Caucia, F., & Marinoni, L. (2010). XRPD patterns of opals: A brief review and new results from recent studies. Powder Diffraction, 25(3), 274-282. DOI: https://doi.org/10.1154/1.3478554. . The presence of cristobalite is not possible in this type of opal. Probably in more diagenitized diatomites with opal C and opal CT, different results would be obtained and therefore the results presented would only apply to samples with opal A, which should be specified in the text.

3. The affectation of NaCl in the porosity of NaCl is well known; this chemical compound is used routinely in the dispersion of clays. This point cannot be considered a discovery of the work, and some already existing references that describe this very common process should be introduced.

4. Figure 11b represents the structure of kaolinite in a very simplified way. Improve drawing so that the layered structure of these phyllosilicates can be appreciated. Authors can use free software like VESTA (K. Momma and F. Izumi, "VESTA 3 for three-dimensional visualization of crystal, volumetric and morphology data," J. Appl. Crystallogr., 44, 1272-1276 2011) to draw such their structure.

I congratulate the authors for the study, which I find very interesting, hoping that the comments may be useful to them.

Author Response

Thanks for your comments. We have made corresponding revisions according to your suggestions. Please see the attachment.

Reviewer 2 Report

Dear authors

Your effort is appreciated, the article is quite well presented, however, more corrections and explanations are needed.

- Please explain, although the Plasticity properties of soil depend on the conditions of polarity and negative charge of clay particles, do Diamote have these properties?

-How important is the subject of the article - Diamotte soil -? Is the volume of this type of soil too much? and is it available everywhere?

-Also, in this research, if possible, the results of the single-point liquid limit method of Casagrande are compared with the usual multi-point method.

-Investigating the effect of different soil sample drying treatments in determining the plastic limit and the effect of this factor in the classification of the respective soil, is important, please discuss.

-Today, these microscopic organisms and their sediments are widely used in the agricultural and medical industries, construction materials, papermaking, war defense agents, etc. Why has it been paid attention to as soil by respected researchers?

-Do the obtained results emphasize the increase of the attraction force between the soil particles by the diatomaceous earth in the soil and the formation of a clot structure?

thanks

Author Response

(The authors gave the same response as above.)

Reviewer 3 Report

Dear Authors

I have reviewed your paper.  I have several comments to polish manuscript:

1.     The abstract should revise to key findings of this study.

2.     Line 25-26 are not clear. Please revise it.

This study provides further understanding of the Atterberg limits of diatomaceous soil and suggestions for its classification.

This sentence is confused so much.

3.     The abstract should add some results of this research.

4.     It is suggested to discuss more about the findings of this study at the end of abstract.

5.     It is recommended to mention about the applications of this study at the end of abstract:
The findings of this study can help for better understanding of …

6.     The introduction need to consider the different studies to related this work.

7.     It is suggested to add a figure in Section 1, which presents the general sketch of the problem under study.

8.     The introduction may contain the background information, motivation for the study, contributions, and the paper organization. A separate section for the review of literature is recommended to be expounded.

9.     The research gap should highlight in introduction.

10.  The author contributions need to highlight in the paper. The main different of this paper and previous studies need to highlight for readers.

11.  What are the advantages and disadvantages of this study? I recommend the authors to highlight this topic.

12.  What are the limitations of this study? I recommend the authors to highlight this topic.

13.  The authors need to revise to highlight the key contributions by showing clearly the improvement of their work and existing studies.

14.   The quality of all the figures should be improved.

15.  Figure 2 need to revise for better explanation

16.  The title for last section should be changed to "Summary and Conclusions".

17.  The conclusion should improve to highlight the key finding of this paper.

18.  It is suggested to add a nomenclature (including alphabetic letters, Greek letters, subscripts, and superscripts).

19.  The paper need to send to native speaker for editing the entire manuscript.

20.  The reference list can be updated.

Author Response

(The authors gave the same response as above.)

Round 2

Reviewer 3 Report

No more comments for this manuscript. The current form can be published